# Research on Continuous Error Compensation of a Sub-Arc-Second Macro/Micro Dual-Drive Rotary System

**DOI:** 10.3390/mi13101662

**Published:** 2022-10-02

**Authors:** Manzhi Yang, Linyue Li, Chuanwei Zhang, Yumei Huang, Hongzhang Wu, Bin Feng

**Affiliations:** 1College of Mechanical Engineering, Xi’an University of Science and Technology, No. 58 Yanta Middle Road, Xi’an 710054, China; 2School of Mechanical and Precision Instrumental Engineering, Xi’an University of Technology, Xi’an 710048, China; 3Xi’an Institute of Metrology, Xi’an 710068, China

**Keywords:** continuous error compensation, macro/micro dual-driver, sub-arc-second, transformation performance

## Abstract

In this paper, a sub-arc-second macro/micro dual-drive rotary system is designed, and the continuous compensation of the system error and its experimental research are completed. First, the macro-drive system is driven by a direct-drive motor, and the micro-drive system uses a piezoelectric ceramic to drive the micro-drive rotary mechanism; the system uses a micro-drive system to compensate the motion error of the macro-drive system, and uses circular grating to feedback the displacement of the macro/micro total output turntable to form a macro/micro dual-drive closed-loop control system. Second, based on the establishment of the system error model, the design of the system’s continuous error compensation scheme is completed. Finally, the positioning accuracy testing of the system, direct error compensation of the macro-drive, manual error compensation of the macro-drive, error compensation performance of the micro-drive part and macro/micro compensation of the system are carried out in the study. The results show that the repeated positioning error and the positioning error of the system are reduced by 78.8% and 95.2%, respectively, after macro/micro compensation. The correctness and effectiveness of the designed system design, error compensation and control method are verified through performance tests, and its positioning accuracy is verified to the sub-arc-second (0.1 arcsecond) level. The research in this paper has important reference value for the development of ultra-precision macro/micro dual-drive technology.

## 1. Introduction

With the continuous improvement of modern industry requirements for the accuracy of mechanical systems, the requirements for machining accuracy and motion accuracy of mechanical systems used in equipment manufacturing are also growing higher [1,2,3,4,5,6]. Ultra-precision machining technology is an important way to realize the processing of precision machinery and equipment, and is widely used in modern weapons and aviation technology. The macro/micro dual-drive technology was first proposed by Sharon, A. in 1984 [7,8]. The technology adopts macro-drive and micro-drive dual-drives, in which the macro-drive provides large-stroke motion for the system, and the micro-drive compensates for the motion error of the macro-drive, realizing large-stroke and high-precision mechanical motion. Since the concept was put forward, it has become one of the hot spots in the field of precision machinery, and it is currently widely used in high-tech fields, such as the aerospace, biomedicine and military industries [9,10,11,12].

Many scholars have carried out research on the macro/micro dual-drive system. For example, Heui, J. P. et al introduced an ultra-precision long-stroke positioning system; under ultra-precision conditions, according to the design principle of flexible hinges, a micro-table driven by a PZT (piezoelectric ceramic) was designed, some of the design parameters were evaluated, and the positioning stroke reached 200 mm with an accuracy of 10 nm, thus obtaining sub-micron positioning accuracy under long-stroke motion [13]. Al-Jodah, A. et al. designed a new type of large-stroke three-degrees-of-freedom (DOF) micro-positioning mechanism; the finite element analysis results showed that the translation workspace of the mechanism was ±2.5 mm, the rotation workspace was 2.4°, and the mechanism had a high first-order natural frequency [14]. with the macro/micro dual-drive platform designed by Shinno, H. et al., the experiments showed that the maximum movement stroke of the platform was 150 mm, and the positioning accuracy was 0.3 nm [15]. Ku, S. S. et al. designed and fabricated a nanometer-precision 3-DOF positioner capable of combining a fast tool servo and motor-driven feedback controller with a tracking resolution of 2 nm [16]. A new 12-DOF macro/micro dual parallel manipulator was presented; the experimental results showed that the system had good performance, the translational precision could reach 15 μm, the rotation precision could reach 0.35′, and the precision of the micro-manipulator was more than ten nanometers [17]. with the macro/micro dual-drive platform designed by Dong, W. et al., the macro-drive used a sonic motor, and the micro-drive used PZT; the experiment result showed that the positioning accuracy of the system could reach 10 nm [18]. With regard to the research status of precision macro/micro dual-drive technology, there are many studies on linear motion macro/micro dual-drive systems, and many mature products are used in practical applications [19,20,21]. However, the rotary motion system is more complex and less researched than the linear motion system. the research on the rotary macro/micro dual-drive technology is relatively backward, which limits the development of the macro/micro dual-drive technology. Therefore, it is particularly necessary to study the macro/micro dual-drive rotary system.

The system error compensation method and the error compensation control method are the necessary guarantees to realize the high precision motion of the system [22,23,24]. Liu et al. studied a thermal error compensation method; based on the analytical calculation, numerical analysis and experimental test of the thermal error of the machine tool spindle, a thermal error model was established and applied to achieve thermal error compensation [25]. Liu et al. designed a macro/micro dual-drive high-acceleration and high-precision positioning platform control system for IC binding; a controller with LQG (linear-quadratic-Gauss) control algorithm combined with feed forward compensation algorithm is derived, and PID algorithm is selected to control the positioning accuracy of the micro-positioning platform [26]. Elfizy, A. T. et al. designed a dual-stage feed drive (DSFD) system and gave a control scheme; the results showed that after compensating the macro-drive system with the micro-drive system, the precise positioning of the system was reduced by 83% compared with the maximum tracking error of a single macro-drive [27]. Yao et al. proposed the On-line Asynchronous Compensation Methods (OACM) for static/quasi-static error caused by thermal deformation and machine geometry; the results of roller guide grinding experiment indicated that this technology could reduce more than 70% of the machining error caused by thermal deformation [28]. An analytical model was proposed to compensate the compliance error of the delta parallel robot as a mini-robot, and its deflection accuracy along the prescribed trajectory was theoretically improved by 82.6% [29]. Hu used embedded technology to design and develop a real-time compensation controller for the thermal error of CNC machine tools. Using the indirect compensation system, the compensation controller could satisfactorily predict and compensate the thermal error of CNC machine tools [30]. A CSA-based readout circuit with quadrature error compensation through a feedback path for vibratory MEMS Gyroscope was designed; the deviation due to the 1-degree phase error of the demodulator was reduced by about 95% [31]. Shen et al. established a dynamic model of road header based on cutting load; the experimental results showed that the coning and sculling error of the strapdown inertial navigation system could be reduced by at least 52.21% and 42.89%, respectively [32]. Li et al. proposed a machining error compensation method for plate parts based on in-machine measurement; according to the program of scaling error compensation, the accuracy and quality of the machine tool are improved by adjusting the rotation matrix and translation vector [33]. However, most of the current studies use single compensation, the compensation effect of which is limited. Continuous error compensation can enable the system to obtain higher positioning accuracy [34,35,36]. Therefore, research on the error compensation of the macro/micro dual-drive system with various compensation methods is of great significance.

Due to the characteristics of the macro/micro dual-drive feed system structure, factors such as macro-drive system, the micro-drive system itself, and the real-time mutual coupling of macro/micro dual-drives cause errors in positioning accuracy and motion. In order to improve its positioning accuracy, the problem of positioning error compensation in a macro/micro dual-drive system is studied has important practical significance. Therefore, this study designed a macro/micro dual-drive slewing system with sub-arc-second precision, completed the system control principle, error modeling and continuous error compensation, and proved the effectiveness and accuracy of the system design through experiments. The experimental results showed that the system can achieve sub-arc-second-level positioning accuracy. The positional accuracy being restricted to the sub-arc-second level means that the maximum position error of the rotary motion system is limited to the scope of the sub- arc-second level. The rotary motion at the sub-arc-second level has essential applications in micromachining fields, such as ultra-precision machining, chip processing and ultra-precision medical instruments. The micro-drive rotary system provides a way to realize sub-arc-second positioning; therefore, a micro-drive rotary system with sub-arc-second positional accuracy has important value in the above micromachining fields. Thus, it is important to study the positioning performance of a sub-arc-second micro-drive rotary system, and this study can promote the development of ultra-precision micromachining technology.

The rest of this paper is organized as follows. Section 2 introduces the research method of this paper, including system design and error modeling, error compensation scheme and experimental setup. Section 3 presents the experimental study of the system error continuous compensation and discusses the experimental results. The conclusions are presented in Section 4.

## 2. Research Methods

### 2.1. System Structure Design

The design of the macro/micro dual-drive precision rotary system directly affects the motion accuracy, response speed and stability of the system. In this paper, the overall design of the macro/micro dual drive precision rotary system is shown in Figure 1, in which the direct drive motor 6 provides the system with a large-stroke macro-drive displacement, and the micro-drive rotary mechanism 11 provides the micro-drive displacement, and performs error compensation for the macro-drive displacement movement to realize the precise large-stroke rotary motion of the system. The micro-drive rotary mechanism 11 is driven by the PZT 12 during operation. The macro-drive and micro-drive of the system are in a series relationship; that is, the macro-drive rotary displacement of the direct drive motor, 6, drives the rotary table, 13 to rotate through the micro-drive rotary mechanism, 11. The cross bearing, 3, is a rotating guide rail, and its moving ring is connected to the rotary table, 13 through the intermediate revolved body in the middle, 9, and the output part of macro-drive; the other five-direction DOF constraints other than the rotation around the z-direction act. The direct drive motor (D-D motor) of Japan YOKOGAWA (Yokogawa Electric) model DM1C-004 was selected for the macro-driver, and P-235.1S PZT was selected for the micro-driver.

In this paper, a symmetrical micro-drive rotary mechanism was designed, and its three-dimensional view is shown in Figure 2. When the micro-drive rotary mechanism is displaced by the force in the working direction in the internal measurement, a certain amount of rotation angle change will be generated at the output end of the system, thereby converting the linear motion of the micro-drive inside the mechanism into the rotary motion of the mechanism rotary system. According to our research group’s previous research paper [37], the micro-rotation mechanism has a high linearity when it carries out straight-rotation motion and does not generate transverse force, and the positioning accuracy can reach sub-arc-second level.

The working-principle diagram of the macro/micro dual-drive precision rotary system is shown in Figure 3. The macro-drive system and the micro-drive system are independent of each other and adopt a series structure. The working principle of the system is as follows: the direct drive motor and its driver realize the closed-loop control of the macro-drive system; the PZT and its driver realize the closed-loop control of the micro-drive system. The precision circular grating detects and feeds back the output of the macro/micro dual-drive rotary system, and the clipper controller automatically assigns the control operation of the motion ratio to the macro-drive system and micro-drive system; the dual-frequency laser interferometer measures the angular displacement of the output platform and detects the motion positioning accuracy of the system, and can be used to manually compensate the system error.

The system consists of the macro-drive part and the micro-drive part in a series structure, and the error caused by the motion of the macro-driving mechanism is compensated by the micro-drive system. the control scheme of the macro/micro dual-drive system is shown in Figure 4. As shown in the figure, the two parts are independent of each other in their respective motions, the entire system is controlled by the clipper controller, the motion error is detected and fed back by the circular grating, and the displacement error is output to compensate for the motion of the system, finally achieving high-precision positioning.

### 2.2. Systematic Error Modeling

The macro/micro dual-drive precision rotary system is aimed at realizing large-stroke and high-precision motion. Among them, the main error source of the system is the positioning error, so it is necessary to compensate for it.

In order to improve the positioning accuracy and reduce the positioning error of the system, two commonly used methods are the error prevention method and the error compensation method. The error prevention method is mainly to improve the structural design, manufacturing, assembly environment and installation accuracy of the components in the system to reduce the error of the system itself, and reduce the original error and the influence of the original error. However, for the high-precision macro/micro dual-drive rotary system, this method has harsh conditions, high cost, and high requirements for process level and engineering technology, so the error compensation method is usually used.

The error compensation method is a method of artificially creating an error equal to the original error and opposite in direction to compensate and correct the original error, to reduce the original positioning error of the positioning system.

According to the previous research of our research group, the error modeling of point-to-point motion and continuous motion are obtained respectively [38] Only a brief description is given here.

In point-to-point motion, the basic principle of positioning error compensation is to artificially create an error equal to and opposite to the original error in order to compensate and correct the original error. The error compensation equation is
(1)$$\theta_{i}\;{+\;\theta }_{i}^{\prime }=0\;(i=1,2,\ldots ,n)$$

Among them, $\theta_{i}$ is the positioning error value generated at each positioning position, $\theta_{i}^{\prime }$ is the error correction value, and n is the number of compensation points.

This motion can be compensated by the pitch compensation method and backlash compensation method, and the maximum compensation modifier is
(2)$$\theta_{\max}^{\prime }=\max\left\{\left|-\theta_{0}\right|\left|-\theta_{1}\right|\ldots \left|-\theta_{i}\right|\ldots \left|-\theta_{n}\right|\right\}$$

In continuous motion, using polynomial fitting method and selecting polynomial fitting model and the known data as $\left(x_{i},y_{i}\right),i=0,1,\ldots ,n$, one can obtain a polynomial not exceeding m (m < n) degree:(3)$$P_{m}\left(x\right)=\overset{m}{\underset{k=0}{\sum }}a_{k}x^{k}$$

Make $\sqrt{\overset{n}{\underset{i=0}{\sum }}{\left[y_{i}-{\;P}_{m}{(x}_{i}^{2})\right]}^{2}}$ obtain the minimum value, that is, $\overset{n}{\underset{i=0}{\sum }}{\left[y_{i}-{\;P}_{m}{(x}_{i}^{2})\right]}^{2}$ minimum.

Obviously, Formula (3) is an error model for continuous motion. The error compensation equation for continuous motion is
(4)$$\theta^{\prime }\left(x\right){=-P}_{m}\left(x\right)=-{\;a}_{0\;}-{\;a}_{1}{x\;-\;a}_{2}x^{2}-{\ldots -\;a}_{m}x^{m}$$

The maximum compensation correction amount is
(5)$$\theta_{\max}^{\prime }=|{-P_{m}{(x}_{i})|}_{\max}=\max\left\{\left|-\theta_{0}\right|\left|-\theta_{1}\right|\ldots \left|-\theta_{i}\right|\ldots \left|-\theta_{n}\right|\right\}$$

### 2.3. Design of Continuous Compensation Scheme for System Error

From the perspective of the motion characteristics of the macro/micro dual-drive, the error compensation for the system is mainly divided into two parts: the compensation for the macro-drive system and the compensation for the macro/micro dual-drive system. Macro-drive compensation includes direct compensation and manual compensation, while macro and micro compensation are mainly based on macro-drive compensation and use the motion of the micro-drive system to perform system motion error compensation for macro-drive.

Because the elongation of PZT is limited, macro-drive compensation should be used to control the system motion error within the micro-drive compensation range before the macro/micro compensation of the system motion error. Therefore, the compensation method for the overall macro/micro dual-drive system comprises direct macro-drive error compensation, macro-drive manual error compensation, and macro/micro compensation, three continuous compensation methods to achieve precise compensation for the system motion error.

Because the direct drive motor selected in this system has its own adjustment and control function, the direct drive motor is used to directly compensate the error generated by macro-drive. The PMAC control system of the clipper multi-axis motion controller controls the motor driver, and then controls the motor. The compensation is based on the principle of automatic interpolation, and the motion error is compensated according to the error compensation table placed in the system. This compensation is simple and can be repeated.

Manual compensation of macro-drive error refers to further compensation for the motion error of the system based on direct compensation of macro-drive error. Its working principle is that if the target motion point of the system is a, and the actual motion position of the system is a’, then the system error ε is
(6)$${\varepsilon =a}^{\prime }-\;a$$

The motion function value a’’ after manual motion compensation for this position point should be
(7)a″ = a − ka′

Among them, the coefficient k in the Formula (7) is the error proportional coefficient, which can be adjusted appropriately according to the size of the actual error to obtain an ideal compensation effect. First, the position error of each measurement point is detected by the laser interferometer, then the compensation value of each position point is calculated by Formula (7), the error compensation model is established according to the error of the measured motion position point and finally the motion pulse value of each point is calculated. This is written into the controller according to the Clipper language, finally realizing precise control of the end of the movement position.

Macro/micro compensation is based on the completion of macro-drive compensation, and further compensates the motion error of the system as the final error compensation of the macro/micro dual-drive rotary system. According to the characteristics of the PZT driver, the rotational angular displacement output by the micro-drive mechanism has a linear relationship with the voltage of the PZT brake, which is
(8)Δθ=aU+b
where Δθ is the output angular displacement, U is the input voltage of the PZT and a and b are coefficients.

Thus, it can be concluded that the input is
(9)$$U=\frac{\Delta \theta }{a}-\;b$$

According to Formula (17), if the angular displacement to be output is known, it can be controlled by controlling the input voltage of the PZT driver to complete the compensation for the angular displacement of the macro-driver output; that is, the macro-driver of the microsystem performs error compensation.

### 2.4. Experiment Setup

In order to reduce the impacts of the vibration, noise and temperature changes on the test, after the installation of macro/micro precision rotary system in the heart of the constant temperature laboratory, it is then fixed on the test box. The experimental box is placed on a vibration isolation platform, and a complete set comprising a laser interferometer temperature compensation table and a humidity compensation table are installed inside at the same time. Severe vibration and other machine vibration interference should be avoided. The main devices and test instruments are: DM1C-004 direct drive motor, P-235.1S PZT, RB1406 cross bearing, A-type φ150 × 130-type circular grating, ML10 dual-frequency laser interferometer and clipper multi-axis control device. An experiment diagram of the macro/micro dual-drive system is shown in Figure 5.

In this experimental environment, five experiments were completed, namely a macrodynamic positioning accuracy detection experiment, a system macro-positioning error compensation experiment, a macro-manual compensation experiment, a system micro-positioning error compensation experiment and a system macro/micro compensation experiment. These experiments are given in detail in Section 3 of this paper.

## 3. Results and Discussion

### 3.1. Macrodynamic Positioning Accuracy Detection Experiment

A target position point was set every 15° in the range of 0°–360° of the system. According to the automatic analysis system in the laser interferometer, the positive direction (0°–360°) plus the negative direction (360°–0°) is one time measurement. In order to ensure the measurement accuracy, in the macro motion system, five positive and negative positioning error measurements were carried out on all points. Therefore, the positive and negative positioning error detection of each point formed five lines, with the result shown in the Figure 6. When the laser interference detection system detected the positioning accuracy of the system, three consecutive measurements of each point were required to obtain the results, so the positive and negative positioning accuracy of the system was represented by three lines (the extra red line is the positive direction and negative positioning accuracy average), as shown in Figure 7.

For the experiments in Section 3.1, Section 3.2, Section 3.3 and Section 3.5, the positive direction refers to 0°–360°, and the negative direction refers to the positive direction 360°–0°. The measurement results of the positioning errors of all points were five positive lines and five negative lines; the measurement results of the positioning accuracy of the system were three positive lines, three negative lines and one average line.

According to the experimental results, the maximum forward positioning error of the system was 10.79″, and the maximum negative positioning error was −26.28″. At this time, the positioning accuracy of the system was consistent with the repeated positioning accuracy (37.44″ and 3.77″), which was consistent with the technical parameters given by the direct drive motor (maximum positioning accuracy is ±20″ and maximum repeated positioning accuracy is ±3″). However, if the system sub-arc-second-level positioning was realized, the system error needed to be controlled within ±3.00″ (determined by the system micro-drive compensation range), so further compensation for the macro-drive error was required.

### 3.2. System Macro-Positioning Error Compensation Experiment

The system macro positioning error compensation experiment was divided into two parts: the macro-drive direct compensation experiment and the macro manual compensation experiment. The study performed automatic direct compensation of errors in the experiment of Section 3.1, drove the motor to run and executed the direct compensation mode, imported the prepared error compensation table into the clipper controller, and used the method in Section 3.1 of this paper to detect the motion positioning accuracy after compensation. Figure 8 shows the motion error of each point of the system macro-drive after direct compensation, and the positioning accuracy after direct compensation of system macro-drive error is shown in Figure 9.

It can be seen from the experimental results that the maximum positioning error of the system in the positive direction was 11.43″, and the maximum positioning error in the negative direction was −10.27″. The positioning error of the system was 40.4% lower than that former compensation, but the motion error of the system still did not reach the micro-drive compensation range. Therefore, a further manual compensation was necessary.

### 3.3. Macro-Manual Compensation Experiment

In the macro-manual compensation experiment, the error table of direct error compensation was converted into the motion error pulse of each point, the error pulse value was subtracted from the position pulse of each point and the manual compensation motion value of the point was calculated. The formula for the point motion value was
(10)$$y_{Z}=\frac{{655,360\;(y}_{M}-\;\varepsilon )}{360\;\times \;3600}$$

In the formula, $y_{Z}$ is the motion value of each point after compensation (number of pulses), $y_{M}$ is the target position value (angle value) of the motion and ε is the motion error value (angle value) of the point.

In accordance with this method, the manual compensation motion values of all measurement points were calculated, and the motion value table of each point was written into the clipper controller. The motion error of each point of the macro-drive of the system after manual compensation is shown in Figure 10, and the positioning accuracy is shown in Figure 11.

The experimental results showed that the maximum forward positioning errors of the system were 2.0″ and the maximum negative positioning errors were −2.68″. Compared with the manual compensation, the positioning errors of the system were reduced by 75.5%, and the maximum errors of the system were controlled within the range of ±3.0″, reaching the range of the system of micromotion compensation.

### 3.4. System Micro-Positioning Error Compensation Experiment

Before macro/micro compensation, the relationship between the working voltage U of the PZT and the rotational deformation Δθ of the micro-rotation mechanism should be measured to calculate the voltage required for micro-error compensation. In this experiment, the deformation of the PZT system was tested in the rising stage (5 V–9.8 V) and the falling stage (5 V–0 V). The relationship between the voltage and the deformation is shown in Figure 12.

The fitted linear equations for the ascending and descending phases are
(11)U=1.4667Δθ+5.1
(12)U=1.5858Δθ+4.8

The linearity is 0.996 and 0.995, respectively, which proves that the linearity of the system in the bidirectional motion stage is good.

### 3.5. System Macro/Micro Compensation Experiment

The macro/micro compensation was performed on the system based on the macro-drive system positioning error compensation. In the process of moving from one target point to the next, the micro-drive input was zero, (the macro-drive moved according to the control program after macro-drive compensation), and the system detected the motion error. When the target point was reached, the macro-drive input was zero (the servo is on), and the micro-drive compensated the system error. The error of each point was calculated by Formulas (11) and (12) and the voltage to be compensated was imported into the control program to realize the micro-drive compensation. After compensation, the laser interferometer was used to conduct the two-way five-times detection system positioning accuracy. The positioning error of each point after compensation is shown in Figure 13.

After macro/micro compensation, the positive repeatability, negative repeat positioning accuracy, positive positioning accuracy, negative positioning accuracy and bidirectional positioning accuracy were: 0.8″, 0.8″, 0.9″, 0.9″ and 1.8″, respectively. The positioning accuracy of the system before and after compensation was greatly improved.

### 3.6. Discussion

In accordance with the experimental conditions, the system macro-drive positioning accuracy detection experiment, the system macro-drive direct compensation experiment, the system macro-drive manual compensation experiment, the system micro-drive compensation performance experiment and the system macro/micro compensation experiment were completed successively. The main analyses of the experiment were as follows:
(1)In the system macro positioning error compensation experiment, the positioning error of the system after the system macro drive direct compensation was 40.4% lower than that before compensation. In the macro manual compensation experiment, the positioning error of the system after the manual compensation of the system macro-drive was reduced by 75.5% compared with that before the manual compensation. The validity and goodness of the two methods of macro-drive error compensations of the system were verified by the experiments.(2)The positioning accuracy parameters before and after system continuous error compensation are shown in Table 1. After the system had been continuously compensated for the positioning error, the positioning accuracy was significantly improved. The repetition error, one-way positioning error, and positioning error reached 0.8″, 0.9″ and 1.8″, which represented reductions of 78.8%, 97.3% and 95.2%, respectively, proving the effectiveness and precision of the positioning error continuous compensation method and the error compensation control of the macro/micro dual-drive precision rotary system.(3)Compared with other studies, this study achieved a higher degree of error compensation, as shown in Table 2. This error compensation method has a better error compensation effect.


The experimental results of the macro-drive system positioning accuracy of the system show that the motion accuracy of the system before compensation is consistent with the technical parameters of the direct drive motor, indicating that the system has a reasonable structure design, excellent assembly performance and good control system adjustment.

## 4. Conclusions

This paper designed and developed a macro/micro dual-drive precision rotary system that can provide large stroke, high precision and full load rotary motion. The system uses the dual-drive of direct drive motor as the macro-driver and PZT as the micro-driver. In addition, a positioning accuracy detection experiment, a system macro-drive error compensation experiment, a system micro-drive compensation performance experiment and a system macro/micro compensation experiment were completed. The main research conclusions are as follows:
(1)A macro/micro dual-drive rotary system was designed that could realize large-stroke and high-precision motion and could solve the contradiction between large-stroke and high-precision rotary motion and positioning.(2)According to the system motion characteristics, the system error modeling and control scheme research were completed, and three continuous compensation schemes were designed, namely direct compensation of macro-drive error, manual compensation of macro-drive error, and macro/micro compensation, to realize the control of motion error of the system. Precision compensation provides a good foundation for system error compensation.(3)The system motion and error continuous compensation performance experiments were completed, and the experimental results showed that the system’s positioning error was reduced from 37.4″ to 1.8″ (95.2% reduction), and the repetition error, one-way positioning error and positioning error were reduced by up to 0.8″, 0.9″. The results verify the correctness and precision of the system design and the continuous error compensation method The system has a positioning accuracy of sub-arc-second level.


Future work should focus on improving the accuracy of the control process and detection procedure.

## Figures and Tables

**Figure 1 micromachines-13-01662-f001:**
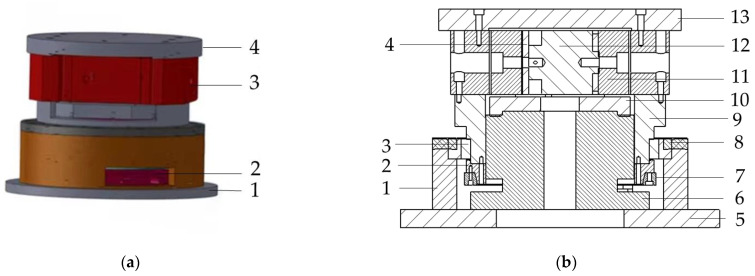
Structure diagram of precision rotary system with macro/micro dual-drive. (**a**) 1, base; 2, direct drive motor; 3, micro rotary mechanism; 4, rotating table. (**b**) 1, bearing shell; 2, inner ring gland of bearing; 3, cross roller bearing; 4, spacer; 5, base; 6, direct drive motor; 7, circular grating encoder; 8, out ring gland of bearing; 9, revolved body in the middle; 10, connecter; 11, micro rotary mechanism; 12, PZT; 13, rotating table.

**Figure 2 micromachines-13-01662-f002:**
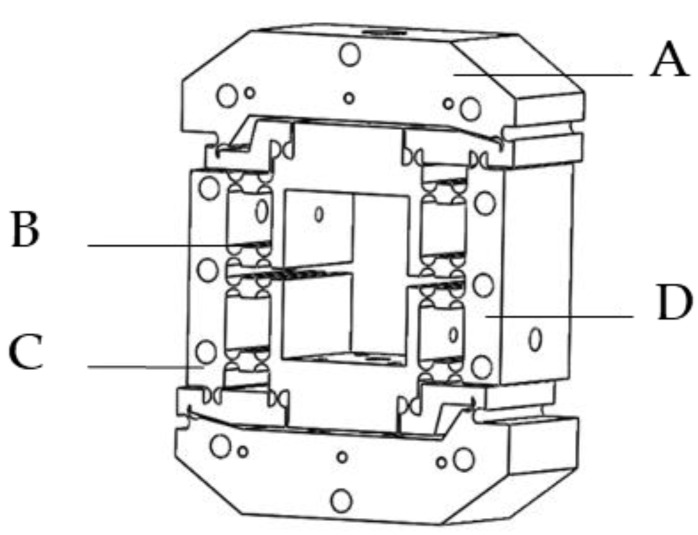
A 3D diagram of the micro rotary mechanism: A, output connection mechanism, B Straight circular flexible hinge mechanism; C PZT micro-drive afferent mechanism; D Macro/micro connection mechanism.

**Figure 3 micromachines-13-01662-f003:**
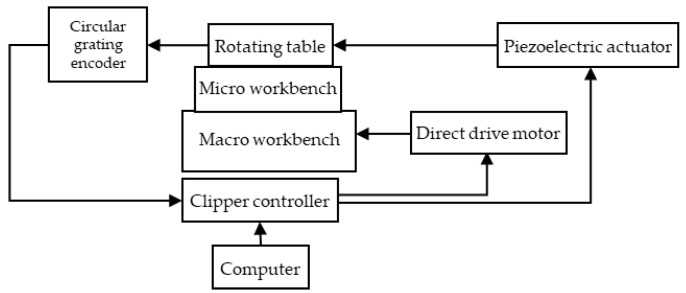
Working-principle diagram of the precision rotary system with macro/micro dual-drive.

**Figure 4 micromachines-13-01662-f004:**
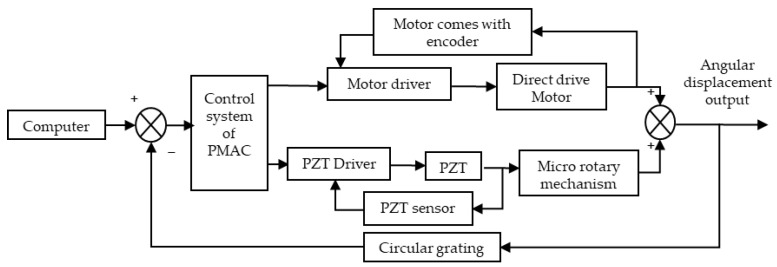
Diagram of control structure of precision rotary system with macro/micro dual-drive.

**Figure 5 micromachines-13-01662-f005:**
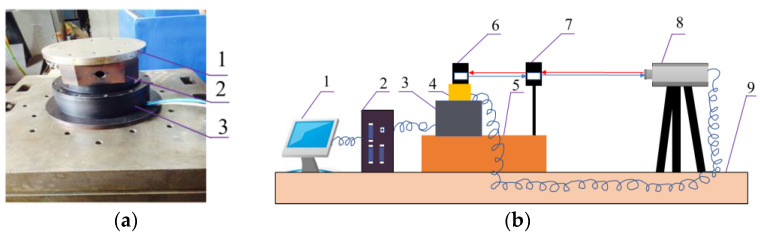
Experiment diagram of macro/micro dual-drive system. (**a**) Picture of macro/micro dual-drive rotary system: 1, rotation workbench; 2, micro-drive system; 3, macro-drive system. (**b**) Experimental program of system: 1, computer; 2, control system of PMAC; 3, macro/micro dual-drive precision rotary system; 4, calibration axis; 5, experimental box; 6, reflex mirror; 7, interference mirror; 8, laser (Renishaw’s dual-frequency laser interferometer ML10); 9, vibration isolation.

**Figure 6 micromachines-13-01662-f006:**
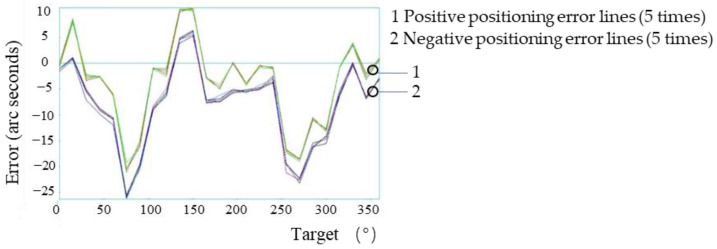
Positioning errors of all points of the system’s macro-drive.

**Figure 7 micromachines-13-01662-f007:**
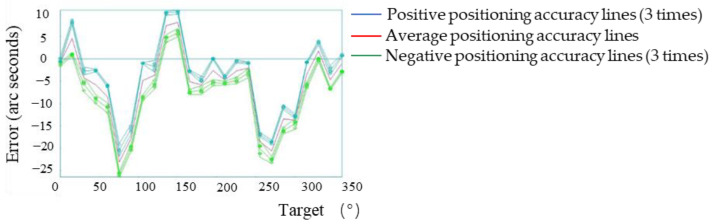
Positioning accuracy of the system’s macro-drive.

**Figure 8 micromachines-13-01662-f008:**
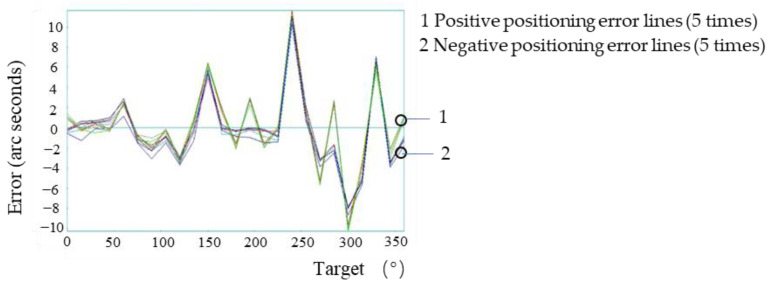
Positioning errors of all points of the system’s macro-drive after direct error compensation.

**Figure 9 micromachines-13-01662-f009:**
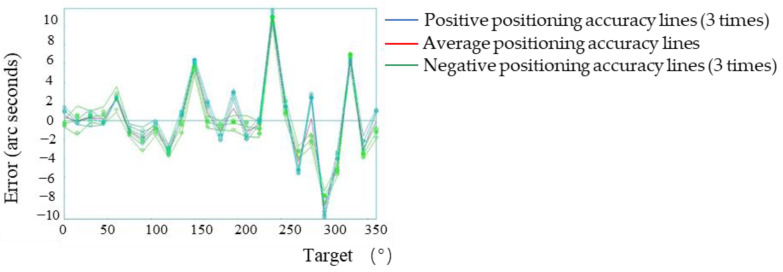
Positioning accuracy of the system’s macro-drive after direct error compensation.

**Figure 10 micromachines-13-01662-f010:**
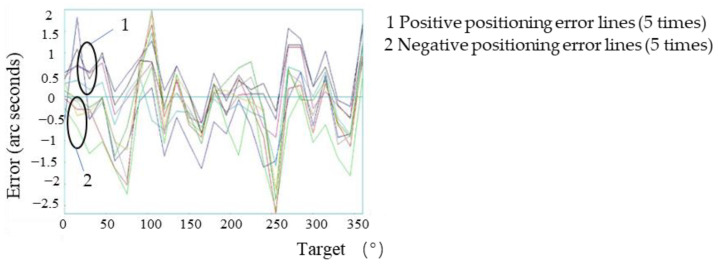
Positioning errors of all points of the system’s macro-drive after manual error compensation.

**Figure 11 micromachines-13-01662-f011:**
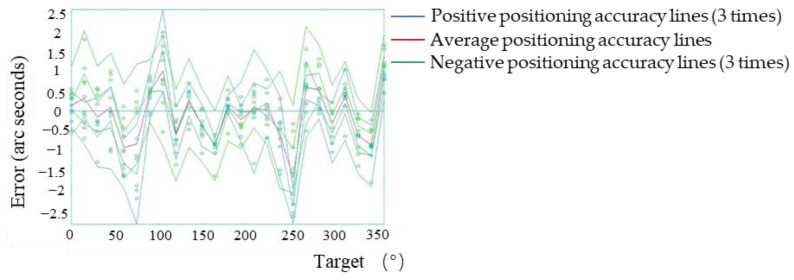
Positioning accuracy of the system’s macro-drive after manual error compensation.

**Figure 12 micromachines-13-01662-f012:**
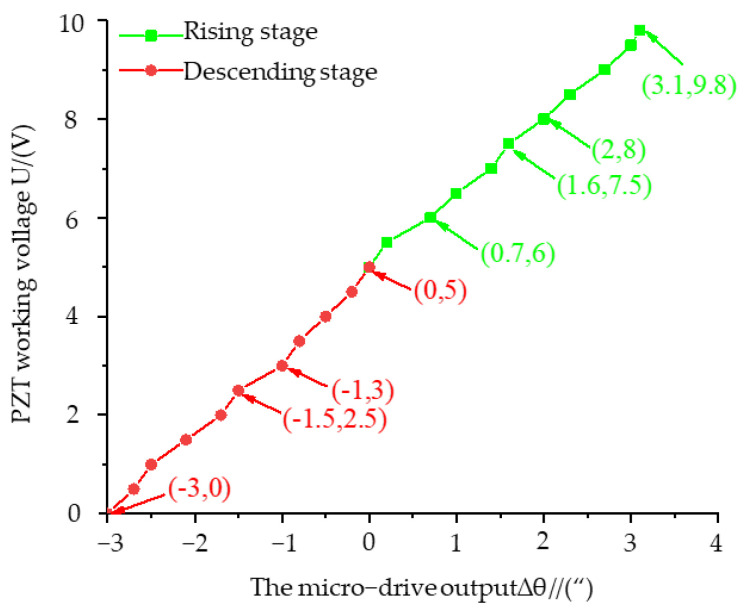
Relationship between the micro-drive output and PZT working voltage.

**Figure 13 micromachines-13-01662-f013:**
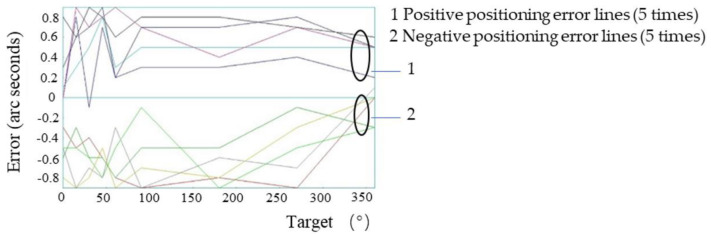
Positioning errors of all points after macro/micro dual-drive error compensation.

**Table 1 micromachines-13-01662-t001:** Main parameters for positioning accuracy of the system before and after macro/micro dual-drive error compensation.

Parameters	Value before Compensation (″)	Value after Compensation (″)	Error Reduction Ratio
Positive repeatability	3.77	0.8	78.8%
Negative repeat positioning accuracy	2.83	0.8	71.7%
Positive positioning accuracy	33.27	0.9	97.3%
Negative positioning accuracy	32.75	0.9	97.3%
Bidirectional positioning accuracy	37.44	1.8	95.2%

**Table 2 micromachines-13-01662-t002:** This design is compared with other references of error compensation.

References	Author	Year	Error Reduction Percentage
[27]	Elfizy A T. et al.	2005	83.0%
[28]	Yao, H. et al.	2012	70.0%
[29]	Nguyen. et al.	2022	82.6%
[31]	Zargari. et al.	2022	95.0%
[32]	Shen, Y. et al.	2021	52.2%
This paper	Yang, M. et al.	2022	95.2%

## Data Availability

The data presented in this study are available on request from the corresponding author.

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
