# Peer review of "Research on Continuous Error Compensation of a Sub-Arc-Second Macro/Micro Dual-Drive Rotary System"

_micromachines, 2022, doi:10.3390/mi13101662_

Round 1

Reviewer 1 Report

This manuscript proposes a continuous error compensation for a self-developed sub-arcsecond macro/micro dual-drive rotary system. The research work is complete and the manuscript is well written. However, there are some issues should be addressed before recommendation for publication.

1)       The ‘Abstract’ section should be more concise.

2)       In the ‘Introduction’ section, the literature review on the macro/micro dual-drive rotary system should be added. Table 4 compares some published works. It is suggested to review and compare them detailed in the ‘Introduction’ section.

3)       Figure 1 is not clear enough to explain the structure of the macro/micro dual-drive rotary system. More views are necessary.

4)       Figure 3, Ø116-0.02 is improperly presented.

5)       In the section ‘5.6 Experimental analysis’, it seems that (3) and (4) have same meaning. Besides, it is recommended to delete the section of ‘5.6 Experimental analysis’, because its main content is same with that of the ‘Conclusions’ section.

Overall, the manuscript presents an experimental work which is more like an engineering practice. The theoretical novelty should be further explained in the revised version.

Author Response

Dear reviewers.

Thank you for your letter dated September 11. We were pleased to know that our work was rated as potentially acceptable for publication in Journal, subject to adequate revision. We thank the reviewers for the time and effort that they have put into reviewing the previous version of the manuscript. Their suggestions have enabled us to improve our work. Based on the instructions provided in your letter, we uploaded the file of the revised manuscript. Accordingly, we have uploaded a copy of the original manuscript with all the changes highlighted by using the track changes mode in MS Word.

Appended to this letter is our point-by-point response to the comments raised by the reviewers. The comments are reproduced and our responses are given directly afterward in a different color (red).

We would like also to thank you for allowing us to resubmit a revised copy of the manuscript.

We hope that the revised manuscript is accepted for publication in the Micromachines.

Sincerely,

Dr. Yang Manzhi

Xi'an University of Science and Technology

Email: xkdymz@xust.edu.cn

Reviewer 2 Report

The article presents the results of research on the topic of the problem of continuous error compensation in a macro-/micro-rotary system with a double drive of less than an arc second. The article may be of some practical interest, but it is absolutely unacceptable in its current form.

As it stands, I cannot recommend the article for publication. A significant revision of the article is required to bring it to the existing standards of scientific articles.

The article is poorly structured, poorly prepared (in particular, the poor quality of most of the Figures), contains a large number of typos. A large amount of well-known information that is redundant and can be significantly reduced.

The abstract is very long, it should be shortened by 1.5 - 2 times. For example, the first two sentences are kind of "Introduction" which is a little weird for an Abstract. The task of the Abstract is to briefly report the main achievements of the article. It needs to be focused on that.

The introduction as a whole is quite well written and fully substantiates the goals and objectives of the work. Some questions are raised by the need for the last section of the Introduction (lines 120-124) - I see no point in prescribing sections of the article.

I recommend using the sequence of sections accepted in the literature: Introduction - Materials and Methods - Results - Discussion - Conclusions. If you use this system of sections, then it turns out that the Materials and Methods section takes up more than half of the article! This is completely redundant. The article (as a rule) mainly describes not the research methods, but the direct results of the research. Research methods are described briefly and in a little more detail only if the methods themselves are original and have not been described before. For standard methods, it is enough to give a brief description and relevant references for a more detailed study.

The use of triple section numbering (1.1.1.) is undesirable.

Fig 1 - the numbering of parts is rather rough - I recommend doing it in a better and more aesthetic way.

Tab 1 - Parameters. There are no spaces between values and their dimensions. (same for tab 2)

It is not very clear why Figure 4 is needed - besides, it is of poor quality. This is a scientific article, not a product catalog.

The first subsection of section 5 also describes the experimental procedure. This should be described in the Materials and Methods section.

Figure 9 provides very little information, and the images are of poor quality. Instead of these low-quality photographs, you can give a scheme for the experiment.

Fig 10 and beyond - what do the lines of different colors mean? This should be indicated in Figure and described in the caption.

All graphs are of poor quality - the font of the axis names does not match the font of the corresponding scale parameters.

The Conclusion should list the main results of the work - preferably by points (1. 2. 3. ...).

References do not meet the requirements of the journal.

Author Response

Dear Reviewers,

Thank you for your letter dated September 11. We were pleased to know that our work was rated as potentially acceptable for publication in Journal, subject to adequate revision. We thank the reviewers for the time and effort that they have put into reviewing the previous version of the manuscript. Their suggestions have enabled us to improve our work. Based on the instructions provided in your letter, we uploaded the file of the revised manuscript. Accordingly, we have uploaded a copy of the original manuscript with all the changes highlighted by using the track changes mode in MS Word.

Appended to this letter is our point-by-point response to the comments raised by the reviewers. The comments are reproduced and our responses are given directly afterward in a different color (red).

We would like also to thank you for allowing us to resubmit a revised copy of the manuscript.

We hope that the revised manuscript is accepted for publication in the Micromachines.

Sincerely,

Dr. Yang Manzhi

Xi'an University of Science and Technology

Email: xkdymz@xust.edu.cn

Round 2

Reviewer 1 Report

It can be accepted in its current form.

Author Response

Dear Editor, Dear reviewers

Thank you for your letter dated September 29. We were pleased to know that our work was rated as potentially acceptable for publication in Journal, subject to adequate revision. We thank the reviewers for the time and effort that they have put into reviewing the previous version of the manuscript. Their suggestions have enabled us to improve our work. Based on the instructions provided in your letter, we uploaded the file of the revised manuscript. Accordingly, we have uploaded a copy of the original manuscript with all the changes highlighted by using the track changes mode in MS Word.

Appended to this letter is our point-by-point response to the comments raised by the reviewers. The comments are reproduced and our responses are given directly afterward in a different color (red).

We would like also to thank you for allowing us to resubmit a revised copy of the manuscript.

We hope that the revised manuscript is accepted for publication in the Micromachines.

Sincerely,

Dr. Yang Manzhi

Xi'an University of Science and Technology

Email: xkdymz@xust.edu.cn

Reviewer 2 Report

Since the authors have done serious work and significantly improved the quality of the manuscript, I can recommend the article for publication. There are a few small recommendations that I advise you to pay attention to:

11.       The abstract is not part of the article. Abbreviations may be introduced into the Abstract (although this is not desirable) only if they are frequently used in the Abstract itself. Thus, I do not recommend introducing the abbreviation piezoelectric ceramic (PZT) in the Abstract, but at the beginning of the Introduction.

22.       As a rule, when specifying the reference, only the name of the first author is indicated. The full name and surname (Professor Andre Sharon of the Massachusetts Institute of Technology or Bijan Shirinzadeh et al from Monash University in Australia) is too long in my opinion. It is also obvious that the University of California is located in the United States. In my opinion, such things can be completely reduced in a scientific article.

33.       Table 2. - it is still necessary to indicate the universal accuracy: 52.21% and 82.6% and 83% - this is incorrect - you either need to increase the number of decimal places (83.00% - if such accuracy is important) or, conversely, reduce it (52%). The same - in the Conclusion "positioning error is reduced from 37.44″ to 1.8″" - either everywhere one or everywhere two decimal places.

Author Response

(The authors gave the same response as above.)
